Structure, ontogeny and evolution of the patellar tendon in emus (Dromaius novaehollandiae) and other palaeognath birds

Regnault Sophie 1 sregnault@rvc.ac.uk
Pitsillides Andrew A. 2
Hutchinson John R. 1
1 Structure and Motion Laboratory, Department of Comparative Biomedical Sciences, The Royal Veterinary College , Hatfield , United Kingdom
2 Department of Comparative Biomedical Sciences, The Royal Veterinary College , London , United Kingdom
Druzinsky Robert
Electronic publication date: 2014 Dec 23
Publication date: 2014
Volume: 2
Electronic Location ID: e711
Received 2014 Jun 4; Accepted 2014 Dec 6
Copyright: © 2014 Regnault et al.
Copyright year: 2014
Copyright holder: Regnault et al.
License: This is an open access article distributed under the terms of the Creative Commons Attribution License, which permits unrestricted use, distribution, reproduction and adaptation in any medium and for any purpose provided that it is properly attributed. For attribution, the original author(s), title, publication source (PeerJ) and either DOI or URL of the article must be cited.
License URL: https://creativecommons.org/licenses/by/4.0/

Keywords: Sesamoid, Ratite, Anatomy, Histology, Biomechanics, Palaeontology, Patella, Palaeognathae

Funding: Leverhulme Trust RPG-2013-108 This research was funded by a grant from the Leverhulme Trust (RPG-2013-108) to John R. Hutchinson and Andrew Pitsillides, and John R. Hutchinson’s role was enabled by a Senior Research Fellowship awarded by the Royal Society Leverhulme Trust in 2012. The funders had no role in study design, data collection and analysis, decision to publish, or preparation of the manuscript.

==============================
The patella (kneecap) exhibits multiple evolutionary origins in birds, mammals, and lizards, and is thought to increase the mechanical advantage of the knee extensor muscles. Despite appreciable interest in the specialized anatomy and locomotion of palaeognathous birds (ratites and relatives), the structure, ontogeny and evolution of the patella in these species remains poorly characterized. Within Palaeognathae, the patella has been reported to be either present, absent, or fused with other bones, but it is unclear how much of this variation is real, erroneous or ontogenetic. Clarification of the patella’s form in palaeognaths would provide insight into the early evolution of the patella in birds, in addition to the specialized locomotion of these species. Findings would also provide new character data of use in resolving the controversial evolutionary relationships of palaeognaths. In this study, we examined the gross and histological anatomy of the emu patellar tendon across several age groups from five weeks to 18 months. We combined these results with our observations and those of others regarding the patella in palaeognaths and their outgroups (both extant and extinct), to reconstruct the evolution of the patella in birds. We found no evidence of an ossified patella in emus, but noted its tendon to have a highly unusual morphology comprising large volumes of adipose tissue contained within a collagenous meshwork. The emu patellar tendon also included increasing amounts of a cartilage-like tissue throughout ontogeny. We speculate that the unusual morphology of the patellar tendon in emus results from assimilation of a peri-articular fat pad, and metaplastic formation of cartilage, both potentially as adaptations to increasing tendon load. We corroborate previous observations of a ‘double patella’ in ostriches, but in contrast to some assertions, we find independent (i.e., unfused) ossified patellae in kiwis and tinamous. Our reconstructions suggest a single evolutionary origin of the patella in birds and that the ancestral patella is likely to have been a composite structure comprising a small ossified portion, lost by some species (e.g., emus, moa) but expanded in others (e.g., ostriches).

Introduction

The patella (kneecap) is a sesamoid; a bone which develops within a tendon passing over a joint (Vickaryous & Olson, 2007). In part because of their anatomical location, sesamoids are thought to increase the mechanical advantage of a muscle by holding its tendon further from the joint and thus lengthening the moment arm of the muscle force (Alexander & Dimery, 1985; Vickaryous & Olson, 2007). Another presumed important role of sesamoids is in tendon protection; allowing the tendon to withstand compression as it bends over joints and acts distantly from the muscle (Sarin et al., 1999; Vickaryous & Olson, 2007). These two functions are the most frequently cited as to why sesamoids like the patella might have evolved and been retained. The patella may confer other functional advantages or develop for other reasons; however, if its morphology when it first evolved differed from its current one (i.e., it may have been exapted into new roles).

The potential sesamoids have for altering the biomechanics of joint motion raises interesting questions about whether they play some small role in the evolution of different locomotor patterns. The patella itself forms within the main extensor tendon of the knee, and is found in many clades of tetrapod vertebrates. In placental mammals, some lizards and birds, the patella is usually ossified (as far as it has been investigated; Haines, 1942; Haines, 1969; Hogg, 1980; Vickaryous & Olson, 2007), whereas in marsupials it is generally fibrocartilaginous (Reese et al., 2001). The avian, lizard (squamatan) and mammalian patellae evolved independently from each other (Dye, 1987; Sarin et al., 1999), but it is unclear when and in what form the avian patella first evolved, and whether it might also have evolved repeatedly within birds. The phylogenetic position of Palaeognathae (ratites and their relatives; emus, ostriches, tinamous, kiwis and kin; Fig. 1) as a relatively basal clade of birds—sister group to the enormously diverse Neognathae, comprising all other crown clade birds—means that they are an important outgroup for exploring the early evolution of traits, such as the patella, in birds. The presence or absence of the patella in palaeognaths gives us clues as to whether it is likely to be an ancestral trait shared by all birds and inherited from a common ancestor, or a derived trait in neognaths and other birds, perhaps convergently evolved multiple times.

Figure 1 Phylogenetic tree showing the major clade division of modern birds into Palaeognathae and Neognathae, and their fossil outgroups.

Phylogeny constructed from the data of Hackett et al. (2008), Harshman et al. (2008), Phillips et al. (2010), Smith, Braun & Kimball (2013) and Mitchell et al. (2014), with outgroup positions from Livezey & Zusi (2007).

The configuration of the main knee extensor tendon in palaeognathous birds is a subject of some ambiguity. Typical tendon in tetrapod vertebrates is composed almost entirely of densely packed collagenous bundles (Schweitzer, Zelzer & Volk, 2010), and this is expected in palaeognaths too. However, some species are reported to have an ossified patella within this tendon (Deeming, 1999; Picasso, 2010), whereas in others the patella is said to be absent (Shanthi et al., 2007); available data and literature indicate several contradictions. The patella’s ability to alter limb joint mechanics means that elucidation of this tendon’s configuration could provide some insight into the evolution of cursoriality and large body size in ratites. Similarity or dissimilarity in knee morphology could also give clues on whether loss of flight and cursorial specialization are likely to have occurred multiple times in palaeognaths (although obviously the inferences that can be made from a single anatomical trait are limited). Skeletal development has long been used as a source of data for phylogenetic analyses (Maisano, 2002; Sánchez-Villagra, 2002; Maxwell & Larsson, 2009), and additionally the little-studied patella may provide another morphological character of use in resolving the unclear and highly controversial evolutionary relationships and history of palaeognathous birds (Cracraft et al., 2004; Harshman et al., 2008; Bourdon, De Ricqles & Cubo, 2009; Johnston, 2011; Smith, Braun & Kimball, 2013; Mitchell et al., 2014).

The incomplete understanding and imprecision regarding form, function, ontogeny and phylogeny of patellae in palaeognaths is typical of most, if not all, sesamoid bones. The frequent perception of patellar structures as unimportant (Pearson & Davin, 1921; Kaiser, 2007; Vickaryous & Olson, 2007) is probably the reason that so few studies have examined this anatomical region in detail, particularly in birds. As such, the configuration of the patella (and its tendon) in palaeognaths is a confused issue. Within the clade, ratites have been described by some to lack the patella (Thompson, 1890; Fowler, 1991; Stewart, 1994; Drenowatz, 1995). In contrast, other studies noted the prominence of the patella in these species (de Vriese, 1909), and an extensive review of avian anatomical characteristics compiled by Livezey & Zusi (2006) lists it as present in all ratites.

Gadow (1880; 1885) writes (though does not illustrate) that ratites do have patellae, but that they give the impression of absence through fusion of the patella to the tibial crest. Superficially this seems supported by other studies, which note fusion of the patella in kiwis and extinct moa (Beale, 1985; Beale, 1991; Sales, 2005; Turvey & Holdaway, 2005). However, Beale (1985) expressed doubt over his identification of the patella in kiwis, noting that such fusion would be unusual. Indeed, Beale’s published radiographs confirm his doubts by showing that the form and position of the ‘patella’ structure is more consistent with that of the cranial cnemial crest of the tibia (also called the crista cnemialis cranialis, proximal tibial ossification centre, or tibial epiphysis).

The cranial cnemial crest is a “traction epiphysis” (Hogg, 1980; Hutchinson, 2002)—an initially separate intratendinous ossification that fuses with the tibia—and so can easily be misidentified as a patellar sesamoid. The true patella remains independent throughout life in most birds and is normally located well proximal to the cranial cnemial crest, within the patellar sulcus of the femur. In contrast, the development of the cranial cnemial crest and its fusion to the tibia (remaining entirely separate from the patella) is well documented in embryological/ontogenetic studies (Hogg, 1980; Pourlis & Antonopoulos, 2013). It seems that kiwis do possess a true independent patella in the expected location, as illustrated in a myological study of adult kiwis (McGowan, 1979), but misidentification seems to be a prominent problem, with several other studies likewise mistakenly identifying the cranial cnemial crest as the patella in birds.

The potential for misidentification of the patella (or cranial cnemial crest) makes it difficult to evaluate statements when there are no clear descriptions or accompanying evidence. Ostriches (Struthio) are well documented to have both a true (unfused) patella and a second more distal ossification, distinct from the first; a so-called double patella (Haughton, 1864; Macalister, 1864; Thompson, 1890; de Vriese, 1909; Deeming, 1999; Gangl et al., 2004; Chadwick et al., 2014). For the other Palaeognathae, however, there are fewer definitive data explicitly exploring patella presence and form, and where they do exist these data lack clear illustrations. For example, some authors refer to a patella in greater rheas (Rhea americana) (Brinkmann, 2010; Picasso, 2010), whilst other earlier literature surprisingly suggests that this species possesses a double patella as in ostriches (Stannius, 1850; de Vriese, 1909). A patella is also referred to in dwarf cassowaries (Casuarius bennetti) (Brinkmann, 2010) but has not been found in a limited study of southern cassowaries (Casuarius casuarius) (Biggs, 2013). Volant tinamous are said to have only a fibrocartilaginous structure in lieu of an ossified patella (Parker, 1864). In extinct elephant birds (Aepyorninithidae), patellar presence has been inferred from muscle scars on the cnemial crest (Livezey & Zusi, 2006). However, the latter evidence is inconclusive because the patella itself would not articulate to such scars, but rather do so via a patellar tendon, and so the scars cannot by themselves establish the presence or absence of a patellar sesamoid bone. An independent patella was mentioned in just one specimen of moa (Dinornithiformes) described by Owen (1883). Thus confusion, misidentification and specimen rarity all combine to make ascertaining the status of the patella in palaeognath species difficult.

Like almost all members of its clade, the presence and configuration of the patella and its tendon in emus (Dromaius novaehollandiae) is also unclear. Myological studies have not explored the question (Patak & Baldwin, 1998), and whilst one study lists patellar presence early in ontogeny (Maxwell & Larsson, 2009), another has described the patella to be absent in adult emu specimens (Shanthi et al., 2007). The former finding, however, is again a cranial cnemial crest rather than a patella (EE Maxwell, pers. comm., 2013). In this study, we first describe the gross and microscopic anatomy of the emu patellar tendon throughout ontogeny, in order to better understand the patellar phenotype in this species. We then put these data into the context of the evolution of Palaeognathae as a whole, using other published data and our own novel observations to reconstruct the evolution of the patella in this clade. In doing so we aim to clarify the status of the patella and its tendon amongst palaeognath species, infer how many times the patella has evolved and its temporal origin(s) within birds, and reconstruct patterns of evolutionary change (i.e., patellar loss or expansion) in this lineage.

Materials and Methods

A note on terminology: the triceps femoris muscle group (e.g., Hutchinson, 2002) is equivalent to the mammalian quadriceps femoris group, and in birds comprises Mm. iliotibialis (3 main heads), femorotibialis (3 heads), and ambiens (2 heads in ostriches; 1 in other palaeognaths). The common tendon of these extensor muscles is called the patellar tendon, aponeurosis, or sometimes ligament. Compositionally, this structure has been shown to be closer to tendon than ligament (Bland & Ashhurst, 1997; Livezey & Zusi, 2006), and it is homologous with the tendon that contains the patella in ratite and other birds, and so herein will be referred to as the patellar tendon, even if a patellar sesamoid ossification is absent.

We harvested the right patellar tendon from nine emu (Dromaius novaehollandiae) cadavers, which were euthanized as part of another study and stored frozen (−20 °C). The time between euthanasia and freezing, and the length of freezing varied between specimens. Slow freezing can damage tissue at a cellular level and result in artefacts (such as discontinuous tissue architecture, cell shrinkage and fluid accumulation) on histological examination and paler staining of tissues with conventional histology stains (Baraibar & Schoning, 1985; Spencer & Bancroft, 2013). Immunohistochemical labelling of tissue antigens is less reliable; however, broad tissue types are still identifiable with standard stains (Baraibar & Schoning, 1985).

The emus used in this study were from a UK farm population (Leicestershire Emus and Rheas, Leicestershire). Four were five weeks old (mean bodyweight 4.41 kg ± standard deviation of 0.94 kg), one was six months (19.3 kg), one was approximately 12 months (30 kg), and three were 18 months (mean 40.2 kg ± 2.25 kg). Emus reach their full body size by about nine months (Goonewardene et al., 2003) and are skeletally mature around one year of age (Goetz et al., 2008), therefore our oldest emus represent full osteological maturity. Most palaeognaths mature at a similarly rapid rate (Turvey, Green & Holdaway, 2005), but by comparison kiwis achieve full skeletal maturity by 5–6 years (Beale, 1991; Bourdon et al., 2009) and some moa would have taken up to ten years (Turvey, Green & Holdaway, 2005). Although not yet a common method in birds, skeletal maturity can also be estimated from the fusion of the cranial cnemial crest (Parsons, 1905; Beale, 1985; Naldo, Samour & Bailey, 1997), which in our emus occurred between about 12 and 18 months (L Lamas & JR Hutchinson, unpublished data, pers. comm., 2014), and was complete in our adult (18 month old) emus. All the animals lacked any obvious musculoskeletal or gait pathologies that might complicate our interpretations.

To gain further data for our meta-analysis, we also examined avian osteological collections (>200 specimens) held by the Natural History Museum (NHMUK; Tring and South Kensington sites for zoological and palaeontological (moa) collections), University Museum of Zoology Cambridge (UMZC), the Grant Museum at the University College of London (LDUCZ), the Queensland Museum (QM) in Brisbane, Australia (ratite skeletons), the Institute of Vertebrate Paleontology and Paleoanthropology (IVPP) in Beijing, China (fossil stem-birds), and Museo Paleontológico Egidio Feruglio (MPEF) in Trelew, Argentina (Rhea/Pterocnemia skeletons). Age and gender of specimens are usually unknown in older museum collections, however all specimens were skeletally mature (as judged by fusion of the cranial cnemial crest). Specimens were visually examined for grossly appreciable bone, and a subset were borrowed for CT scanning diagnosis.

We were also able to sample the right patellar tendon and patellae from a mature (71.3 kg) ostrich (Struthio camelus) and the right patellar tendon of a mature (26 kg) Southern cassowary (Casuarius casuarius), both of which were stored frozen, and the right patellar tendon and patella from a fresh adult (1.8 kg) guineafowl (Numida meleagris). We were only able to access one animal for each of these histological samples; however, the gross morphology of each was checked against museum specimens of adults. These included: 9 Struthio skeletons and 3 cadavers as in Chadwick et al. (2014); 8 Numida from RVC frozen cadaveric specimens; total of 19 Casuarius skeletons, sourced from the Queensland Museum, Brisbane, Australia: QM specimen numbers 03454, 12657, 23518, 30059, 31138 and 31352; also NHMUK specimens: of C. bennetti 1864.7.2.2, 1877.1.27.2, 1909.12.11.1 and of C. casuarius 1852.12.5.20, 1877.1.27.1, 1877.1.27.3, 1899.11.10.1, 1905.11.30.2, S/1952.2.1.14, S/1972.1.9, S/1979.37.5, S/1979.39.1 and S/2010.1.24) and against literature to ensure as much as possible that they were representative of these species (in terms of presence/absence of a bony patella and general morphology). Considering that our data are the most limited (based on negative evidence largely from skeletal specimens) for cassowaries, our inferences about their patellar tendon/sesamoid tissue structure are preliminary.

Computed tomography (CT) scans of the wet specimens (emus, ostrich, cassowary, guineafowl) had previously been taken, using various settings optimized for the visualization of bone. We used these scans to diagnose the presence/absence of osseous patellae within the patellar tendon in situ, and to test for skeletal maturity (fusion of cranial cnemial crest) for each animal.

The patellar tendons were dissected out of the thawed cadavers and fixed in 10% neutral buffered formalin. Samples containing grossly observable bone (i.e., ostrich and guineafowl) were decalcified in 5% formic acid solution. All specimens were sectioned in the sagittal plane along the midline and, size permitting, most were further sectioned in sagittal and transverse directions. The tendon sections were dehydrated and embedded in paraffin wax blocks for histological examination under light microscopy. Microtome sections were cut between 4 and 6 µm, and initially stained with routine Haematoxylin and Eosin. Further to this, the following special stains were used: picro-sirius red, for specific identification of collagen; toluidine blue, for identification of collagen, chondrocytes and metachromasia; SafraninO/Fast green stain for cartilage; and von Kossa for calcium salts/mineralisation (von Kossa, 1901). Ostrich and guineafowl sections were additionally stained with Masson’s trichrome to highlight features of the patellar bone microanatomy (which the emus and cassowary lacked as diagnosed by CT).

We reconstructed evolutionary patterns using the parsimony algorithm in Mesquite software (Maddison & Maddison, 2011), over the phylogeny of Livezey & Zusi (2007) for the main topology of Ornithurae (crown group birds and their closest fossil outgroups). However, we adopted the alternative phylogeny for crown group birds (Neornithes/Aves) found by others (Hackett et al., 2008; Harshman et al., 2008; Phillips et al., 2010; Smith, Braun & Kimball, 2013; Baker et al., 2014; Mitchell et al., 2014), which places tinamous within ratites and ostriches as occupying the most basal phylogenetic position within Palaeognathae. Alternate topologies were also examined in a sensitivity analysis of our conclusions, by comparing the evolutionary patterns of the patella resulting from the enforcement of different phylogenetic frameworks. We specifically examined the topologies of Lee, Feinstein & Cracraft (1997), Cooper et al. (2001), Bourdon et al. (2009); Bourdon, De Ricqles & Cubo (2009) and Johnston (2011).

On this basis, we scored the patella as: absent (0), a small flake of bone (1), a rounded nodule (2), a subrectangular block with strong articular surfaces for the patellar sulcus of the femur (3) or a craniocaudally expanded triangular crest, rising high above the patellar sulcus (4). These five graded scores were assumed to be ordered because they form a continuous series (Table 1). Due to easy confusion between the patella and cranial cnemial crest, we ensured that the patella was only counted as present when there was documented visual evidence of it, rather than if its presence was solely supported by in-text statements (unless these were clear and detailed). This meta-analysis would ideally use multiple specimens (which we did where feasible), and an absent score in juvenile specimens was treated more as uncertainty, rather than as definitive evidence of absence.

Table 1 Character state scores for patellar form in Palaeognathae, Neognathae and Hesperornithiformes, following the scores 0–4 described in the Methods.

Basal outgroups to these taxa (e.g., Apsaravis, Yixianornis, Yanornis, Enantiornithes, other extinct birds) lack a patella (Hou, 1997; Clarke & Norell, 2002; Clarke, Zhou & Zhang, 2006; Zhou, Clarke & Zhang, 2008; Wang et al., 2013). Scores for taxa above the genus level were gauged by comparisons of multiple taxa within that clade and the polarity of characters within it (using published phylogenies as cited in the main text), where variation existed.

Taxon	Patellar form (character state)	
Hesperornithiformes	Proximodistally elongate crest (4)	
Neoaves	Small flake of bone (1)	
Galloanserae	Rounded nodule (2)	
Struthio (ostriches)	Large rectangular block with articular surfaces mediolaterally (3)	
Rheiidae (rheas)	No data (?)	
Tinamiformes (tinamous)	Small flake of bone (1)	
Casuarius (cassowaries)	Absent (0)	
Dromaius (emus)	Absent (0)	
Dinornithiformes (moas)	Absent (0)	
Apteryx (kiwis)	Small flake of bone (1)	
Aepyornithiformes (elephant birds)	No data (?)	

Results

Patellar tendon morphology in ageing emus reveals the lack of an ossified patella

Proximally, the patellar tendon in all ages (five weeks to 18 months) of emus is attached to the triceps femoris muscles via an abrupt and distinctive junction (Figs. 2A and 2B). It is off-white and fairly firm, particularly proximally, close to its muscle junction. Other muscles overlie and are attached to the patellar tendon superficially, but are not ideally positioned, or sufficiently strongly connected, to contribute as much force along their line of action, as is the triceps femoris. The patellar tendon sits within the patellar sulcus on the distal femur, bordered by the lateral (larger) and medial (smaller) femoral condyles. At a position approximately one-third along its length, whilst it is still within the patellar groove, the tendon partially splits to form superficial and deep portions (Figs. 2A–2C). The superficial portion bridges the femorotibial joint and inserts onto the lateral and cranial cnemial crests of the tibiotarsus, whilst the deep portion blends with a grossly fatty structure within the patellar sulcus of the femur, and is also attached to the cartilaginous femorotibial menisci (Fig. 2A); mostly to the medial meniscus, but also to cranial and medial parts of the lateral meniscus. Examination of transverse cross-sections of the patella in emus of all ages (five weeks to 18 months) revealed that splitting of the distal patellar tendon creates a superficial roughly triangular portion and deep, rounded portion (Fig. 2C).

Figure 2 Emu patellar tendon.

(A) Schematic drawing of the emu patellar tendon, shown reflected from the femur. Distally the tendon splits into superficial (sup) and deep (deep) portions, with the former inserting onto the cranial tibial crest while the latter blends with a triangular fatty structure (pad) that attaches to the menisci (men). (B) Excised patellar tendon of an emu in sagittal cross-section, showing the triceps femoris muscle group attaching proximally (tri), overlying muscles superficially (over), and the tendon partially splitting distally to form superficial (sup) and deep (deep) portions. (C) Excised patellar tendon of an emu in transverse cross section. (D) Histological section of the patellar tendon from an 18 month old emu, stained with von Kossa. There was no evidence of calcium salts, which form as a black precipitate, as can be seen in the control tissue (inset).

There was no mineralisation/ossification apparent in dissections of the patellar tendons, and this was also observed microscopically using von Kossa staining of histological tendon sections, which showed a negative result for presence of calcium salts (Fig. 2D). CT scans of the patellar tendons in situ confirmed that there was no patellar sesamoid in any of the emus, and in fact the patellar tendon region is markedly radiolucent with respect to the surrounding soft tissues (Fig. 3A).

Figure 3 CT images of the patellar region in select Palaeognathae.

(A) Emu (Dromaius novaehollandiae) distal femur in transverse section, with the lateral femoral condyle on the left. There is no mineralised patellar structure, and the patellar tendon itself is radiolucent with regard to surrounding soft tissues. The soft tissue opacity near the lateral condyle (∗) represents the m. tibialis cranialis muscle origin. (B) Southern cassowary (Casuarius casuarius) distal femur in transverse section, with the lateral femoral condyle on the left. There is no mineralised patellar structure, and like the emu, the patellar tendon itself is slightly radiolucent with regard to surrounding soft tissues. The soft tissue opacity near the lateral condyle (∗) represents the m. tibialis cranialis muscle origin. (C) Ostrich (Struthio camelus) distal femur in transverse cross section, with bony outcrop which becomes the lateral femoral condyle on the left (black arrow). The ossified proximal patella (white arrow) overlies the region of the lateral condyle and patellar sulcus. (D) Red-winged tinamou (Rhynchotus rufescens specimen NHMUK 1851.11.10.43) proximal tibiotarsus in sagittal cross-section, with the mineralised patella (white arrow) attached by a slip of desiccated patellar tendon, as can be seen grossly (photo insert).

Cartilage-like tissue develops within a predominantly adipose patellar tendon during emu ontogeny

Histological examination of the patellar tendon of emus revealed a basic structure maintained across all the ages examined, with some variations throughout ontogeny. Histology confirmed the presence of an abrupt junction between the patellar tendon and the triceps femoris muscles proximally. In all ages, the collagen fibres here (identified by toluidine blue and picro-sirius red staining) have a dense and predominantly transverse orientation, which become more longitudinally oriented as they penetrate into the tendon distally (Fig. 4). Surprisingly, the body of the patella tendon is overwhelmingly comprised of adipocytes, with collagen fibre bundles running throughout in mixed orientations, resulting in a meshwork-like appearance. In some sections, a blood vessel is seen in the proximal deep region of the tendon, within the adipocyte-rich tissue. The body of adipose/collagenous tissue is bounded (both on superficial and deep surfaces) by continuous, longitudinally-oriented and more classically tendon-like dense collagen bundles (Fig. 5A). Cells in the collagen fibre bundles are typically slender and fusiform with elongate nuclei, resembling tenocytes (Fig. 5B).

Figure 4 Junction between the triceps muscle group (tri) and collagen fibres (col) of the patellar tendon in an 18 month old emu.

(A) H&E. (B) Toluidine blue. (C) Picrosirius red. (D) Picrosirius red under polarised light, with the collagen fibres exhibiting birefringence.

Figure 5 Some histological features of the patellar tendon in emus.

(A) Patellar tendon of an 18 month old emu. Collagen bundles (col) run along the superficial surface (right of image), interspersed with basophilic vesicular and chondroid tissue (unfilled arrows). The body of the tendon is mostly composed of adipocytes (a) and collagen bundles (col) in mixed orientations. Vesicular/chondroid tissue, when present in the body of the tendon, is associated with collagen fibre bundles. H&E. (B) Slender tenocytes within the crimped collagen fibre bundles. H&E. (C) Patellar tendon of a five week old emu, displaying a synovial villus on the deep surface, close to the sagittal midline of the tendon. H&E. (D) Patellar tendon of an 18 month old emu. The edges of the superficial (sup) and deep (deep) portions of the distal tendon have a lining layer of cells (filled arrows). H&E.

Throughout all observed ontogenetic stages, the tendon’s deep surface is covered with flattened or polygonal cells in a layer 2–3 cells thick; these likely are synovial-lining cells. Blood vessels are visible in the synovial sub-intima and deeper fascia, particularly laterally (apparently away from the point of tendon contact with bone). Synovial villi are visible on the deep surface in some sections (Fig. 5C).

Where the tendon splits into superficial and deep portions distally, its surfaces have a lining layer of cells that oppose each other (Fig. 5D). The synovial layer is continuous over the tendon split but does not interdigitate with it.

Histological sections from the five week old emus’ patellar tendons are generally more cellular compared to the older (six, 12 and 18 month old) emus. A further difference between the age groups is the presence of an apparent third tissue type, in addition to the adipose and collagenous tissues already described. In the five week old emu patellar tendons, this tissue is highly vacuolated or vesicular, basophilic with H&E, red with Safranin O/fast green, and metachromatic with toluidine blue stain (Figs. 6A, 6C and 6D). This material is most abundant in the deep, proximal region of the tendon and also close to the tendon split laterally. There are lesser amounts along the deep surface distally, and around the dense collagen bundles running along the proximal (triceps) and superficial muscle-tendon junctions. The substance is also present in minimal amounts within the body of the tendon, especially in the deepest third, where it seems to follow collagen bundles as they travel through the adipose tissue. The cells associated with this material have plump and sometimes finely stippled nuclei, in contrast to the compressed nuclei of adipocytes or the elongate nuclei of tenocytes.

Figure 6 Appearance of cartilage-like tissue within the emu patellar tendon at five weeks (left) and 18 months (right) with different histological stains.

(A) Five weeks with H&E. The tissue is basophilic and vesicular in appearance, often seen in association with collagen fibre bundles. (B) 18 months with H&E, from a region near the proximal muscle junction. The tissue has a more homogenous and less vesicular appearance. Chondrocyte-like cells (filled arrows) sit within the normally eosinophilic collagen fibre bundles, creating an intense localised basophilia. (C) Five weeks with Safranin O/fast green. (D) 18 months with Safranin O/fast green. (E) Five weeks with toluidine blue. (F) 18 months with toluidine blue. Nearby collagen fibres bundles near the chondrocyte-like cells show some metachromatic (purple) staining.

In the patellar tendon of the six month old emu, the vesicular tissue is still present, distributed near the proximal and superficial muscle junctions, close to the deep surface proximally and also laterally prior to the tendon split. None of this vesicular tissue is visible in any appreciable quantity in the tendon distally. This material is less vesicular and more homogenous than in the five week old emus. Certain collagen fibres near to this material also appear slightly more basophilic with H&E and slightly metachromatic with toluidine blue.

Much like in the six month old emu, the vesicular material in the 12 month emu patellar tendon looks less vesicular and more homogenous. In addition, there are some areas where rounded cells sit within chondrocyte-like lacunae, giving the tissue a chondroid (cartilage-like) appearance. The tissue has a similar distribution as in the six month old emu.

In the 18 month old emus, the chondroid and vesicular tissue shows the same features as in the six and 12 month olds (basophilic/metachromatic staining of nearby collagen fibres and chondrocyte-like cells; see Figs. 6B and 6F), but is more developed. This chondroid tissue is abundant along the proximal muscle junction, interspersed between collagen bundles and sometimes contains large clusters of adipocytes. The rounded, chondrocyte-like cells occupying lacunae within the tissue are more numerous and are also present within nearby collagen fibre bundles (Figs. 6B, 6D and 6F). The tissue’s basophilia with H&E, metachromasia with toluidine blue, and red staining with Safranin O is more intense in the 18 month old emus compared to the five week old emus, and these stains served to clearly differentiate the vesicular and chondroid tissue from nearby collagen fibre bundles (Fig. 6).

The patellar morphology in other palaeognaths and outgroups

Like the emus, the Southern cassowary has no osseous patella evident on CT scans, and a mild radiolucency of the patellar tendon region with respect to the surrounding soft tissues (Fig. 3B). Gross morphology and attachments are also similar to the emu: a fatty tendon with superficial and deep portions, the former inserting on the cranial and lateral crests of the tibiotarsus and the latter attaching to the menisci. Though generally fatty, a fibrous area is grossly evident in the proximal and deep region of the tendon (Fig. 7A).

Figure 7 Patellar tendon of a Southern cassowary (Casuarius casuarius).

(A) Cut in sagittal cross-section. The cassowary patellar tendon shows broadly similar anatomy to the emu patellar tendon in Fig. 2B. Most of the tendon appears grossly fatty, but there is a fibrous area in the proximal deep region (black arrow) composed of collagen fibre bundles. The black box indicates the approximate histological region shown in Fig. 7B (not to scale). (B) Proximal region of the cassowary tendon, near the junction with the triceps femoris muscle group, showing collagen fibre bundles (col), adipocytes (a) and metachromatic vesicular/cartilage-like tissue with chondrocyte-like cells (unfilled arrows).

Histological appearance of the patellar tendon in the cassowary is also similar to the emus; predominantly fatty and collagenous, with vesicular/chondroid tissue (particularly in the proximal and deep region; Fig. 7B). The fibrous area seen in the proximal deep region contains less adipose tissue than the rest of the patellar tendon; instead, this area is comprised largely of dense collagen fibre bundles.

Patellar tendon sections from the ostrich and guineafowl are, by comparison, more conventional for birds, with the patellar tendon comprising predominantly dense collagenous bundles and containing ossified patellar sesamoids (one in guineafowl and two in ostriches) (Fig. 8). Similar to our emus and Southern cassowary, the patellar tendon of the ostrich includes regions of cartilage-like tissue, especially between and deep to the ossified sesamoids. It also contains areas with apparently discrete strata of adipose tissue, unlike the diffuse and abundant incorporation of fat within the emu and cassowary patellar tendons. Both the proximal and distal ostrich patellae are almost entirely trabecular bone surrounded by fatty and cellular marrow, with regions of well differentiated cortical bone, mostly superficially, containing secondary osteons running along the bones’ longitudinal axes (Fig. 8C). There are small areas of calcified cartilage at some edges of the bones, but no articular cartilage lining the deep surface as in the guineafowl. The guineafowl patella has an outer compact bone cortex with some longitudinal secondary osteons, inner trabecular structure with mostly fatty marrow, and a thick articular cartilage pad lining its deep surface (Fig. 8D).

Figure 8 Configuration of ostrich and guineafowl patellae.

(A) Part of a longitudinal section of the patellar tendon of an adult ostrich, predominantly composed of thick collagenous bundles with patchy regions of cartilage-like tissue and the occasional solitary adipocyte. H&E. (B) 3D segmentation of computed tomography (CT) scan data from ostrich left knee, adapted with permission from Chadwick et al. (2014). The proximal patella (1) partly overlies the lateral condyle and patellar sulcus of the femur, whilst the distal patella (2) sits close to the cranial cnemial crest of the tibiotarsus (crest). The dotted line shows the cut through the proximal patella seen in Fig 8C. (C) Transverse section of the proximal patella of an adult ostrich, showing osteons (black arrows) in the superficial aspect of the bone, trabeculae and medullary cavity occupying the majority of the patella (m), and calcified cartilage (cc). Superficial, top right of image; medial, right of image. Masson’s trichrome. (D) Sagittal cross-section section of an adult guineafowl patella, showing the triceps muscle group attaching proximally (tri), suprapatellar (sp) and infrapatellar (ip) fat pads, outer cortical bone (cb), inner trabeculated structure, and deep articular cartilage lining (art). Proximal, right of image; deep, bottom of image. H&E.

Based upon many extremely well-preserved complete skeletons, particularly from Cretaceous fossil deposits in China and Mongolia, a patella is clearly absent in extinct birds that are basal to Hesperornithiformes (Table 1) (Hou, 1997; Clarke & Norell, 2002; Clarke, Zhou & Zhang, 2006; Zhou, Clarke & Zhang, 2008; Wang et al., 2013). Although a patella may have originated earlier than in the common ancestor of Hesperornithiformes and Neornithes (see below), the fossil record of many ornithurine taxa argues strongly against this possibility. Hou (1997) speculated that a sliver of bone in a Confuciusornis specimen was a patella, but its disarticulated position renders its identification uncertain, and the absence of a patella in dozens of other Confuciusornithidae we have examined and in the literature casts additional doubt on this identification, which could be pathology or another unusual ossification.

In our studies of museum specimens, we observe a true mineralised patella in multiple tinamou and kiwi specimens (and confirm with CT on a subset of loaned specimens; see Fig. 3D). Where present in these taxa, the patella is a small flake-like bone enclosed in the patellar tendon, situated in the patellar groove of the femur. We also observe the double patella in ostriches, evident in over 12 specimens from diverse collections and well documented in the literature (Haughton, 1864; Macalister, 1864; Thompson, 1890; de Vriese, 1909; Deeming, 1999; Gangl et al., 2004). In contrast, we do not find evidence of patellae in other palaeognath specimens (emus, rheas, cassowaries, moa, and elephant birds).

Our reconstructions of the evolution of the patella in birds show it to have evolved only once in this lineage, in the common ancestor of Neornithes and Hesperornithiformes (Fig. 9). Other trees (not shown) using variations on the two main topologies presented give similar results. No matter how the relationships within Palaeognathae are organised (based on the most detailed recent analyses), our phylogenetic optimizations always show a small flake of bone as plesiomorphic for Neornithes (ancestral for the Hesperornithiformes + Neornithes clade); a condition inherited by the common ancestor of Neognathae. However, patterns of patellar evolution within Palaeognathae itself can vary considerably depending on tree topology.

Figure 9 Reconstructed patterns of patellar evolution.

Phylogenetic trees (as per Fig. 1) constructed from the data of (A) Hackett et al. (2008), Harshman et al. (2008), Phillips et al. (2010), Smith, Braun & Kimball (2013), and Mitchell et al. (2014) and (B) Livezey & Zusi (2007). When the evolution of the patella is reconstructed over these trees and those of other authors, the patella shows a single origin in birds (blue). Key: grey, ?/no data or ambiguous data; white, 0/absent; blue, 1/small flake of bone; green, 2/nodule of bone; yellow, 3/subrectangular block; black, 4/expanded triangular crest.

Discussion

The patellar tendon of emus has a highly unusual morphology, and lacks any evidence of a patellar sesamoid ossification. Studies of ontogeny in other birds suggest that a patella is expected to form shortly after the appearance of the cranial cnemial crest ossification centre (Hogg, 1980; Pourlis & Antonopoulos, 2013). In emus, this ossification centre is already present at hatching (Maxwell & Larsson, 2009) and the failure of a patella to appear over the subsequent 18 months, past the point of cranial cnemial crest fusion and skeletal maturity, indicates that emus do not develop osseous patellae. However, examination of older emus (which our study unfortunately lacks, except for >20 museum specimens studied that are of uncertain but adult ages) would be a useful final test.

Unlike the typical patellar tendon of most tetrapod vertebrates, in which the microanatomy consists predominantly of dense, parallel collagenous bundles (Khan et al., 1999; see also Fig. 8A), the patellar tendon of emus is composed almost entirely of adipocytes contained in a collagen bundle meshwork. The high fat content of the patellar tendon can even be seen on CT scans as a radiolucent region (Figs. 3A and 3B) corresponding to the cross-sectional shape of the patellar tendon (Fig. 2C). Additionally, in older emus there are patchy regions of cartilage-like tissue that are most prominent near the proximal muscle-tendon junction, which corresponds to the location of the patellar sesamoid in most other birds, including the proximal patella in ostriches. This may represent a vestigial, totally unossified sesamoid remnant.

As it first appears, the vesicular tissue seen in the tendons resembles tendon/ligament changes described elsewhere in the literature as mucoid degeneration, myxoid degeneration, chondroid metaplasia, and/or fibrocartilage metaplasia (Khan et al., 1999; Vigorita, 2008; Hashimoto, Nobuhara & Hamada, 2002; Buck et al., 2009). Terminology varies, but characteristically the tendon cells become rounded and chondroid in appearance and produce a stainable, sometimes granulomatous matrix (Järvinen et al., 1997; Buck et al., 2009). This type of tendon change is usually reported in the context of tendon pathology, particularly in cases where the duration of symptoms is longstanding (Hashimoto, Nobuhara & Hamada, 2002). Chondroid cells can also be seen at tendon/ligament junctions with bone (Vigorita, 2008).

Similarly, the presence of fat within tendon is associated with age-related degeneration (Hashimoto, Nobuhara & Hamada, 2002). Accumulation of adipocytes in a tendon, variably called tendolipomatosis, lipoid degeneration, or fatty infiltration, is well documented (Hashimoto, Nobuhara & Hamada, 2002; Gagliano et al., 2013), though the aetiology is unclear. The adipocytes occupy spaces between the collagen fibres and may amass to such a point that they weaken and disrupt the fibres (Józsa, Réffy & Bálint, 1984; Hashimoto, Nobuhara & Hamada, 2002). However, it seems unlikely that the appearance of the patellar tendon in these emus is due to an underlying pathology. The appearance is present from a young age and consistent between individuals, suggesting that it represents a ‘normal’ phenotype for this tendon in emus. Tendinopathies also predispose the tendon to rupture (Sharma & Maffulli, 2005), and there are no reports of patellar tendon rupture in emus. Rather than tendinopathy, this emu patellar tendon composition is more likely indication of a limited ability for tendon remodelling (in this case physiologically rather than pathologically) (Khan et al., 1999). It may also reflect the strange evolutionary history of emus (and other ratites) which seemingly involves loss of the patella, possibly during the convergent evolution of flightlessness (e.g., Harshman et al., 2008; Johnston, 2011; Smith, Braun & Kimball, 2013; Baker et al., 2014).

Tendon is known to modify its structure in response to external mechanical load, and this is a more plausible explanation for the unusual appearance of the emu patella tendon. The presence of a cartilage-like tissue in the adult emus is consistent with fibrocartilage formation observed in tendons subject to bending or compression (Vogel & Peters, 2005). This hypothesis is supported by the apparent ageing-related differentiation of this tissue. In the youngest emus, corresponding regions exhibit vesicular material and rounded cells with plump nuclei; Safranin O-positive staining supports the presence of high proteoglycan content in this tissue. The evidence suggests that the vesicular material may be the extra-fibrillar matrix of the tendon, which contains proteoglycan and has a vacuolated appearance when newly formed (O’Brien, 2005). The vesicular appearance might also result if this tissue derives from metaplastic adipocytes.

In older emus, this apparent extra-fibrillar matrix becomes more homogenously-staining and less vesicular. Some of the rounded cells are surrounded by a lacunar space, giving the impression of chondrocytes. Nearby collagen bundles are also more basophilic (with H&E) and metachromatic (with toluidine blue) and contain chondrocyte-like cells, suggesting they are either transforming or being replaced by this tissue. In the oldest emus, this tissue is highly reminiscent of cartilage, and Safranin O staining indicates it has high proteoglycan content. Regardless, this tissue is far from ossified, and the absence of any documented patellar ossification in the literature on emu morphology at least circumstantially supports the inference that this absence represents the general condition for emus.

The appearance of so much fat tissue within what is supposed to be a tendon is more difficult to understand. A commonly presumed function of sesamoids is protective; reinforcing the tendon from high compressive forces as it bends around bone. Fat pads near joints are hypothesised to have a similar function in cushioning tendon and stress dissipation (Benjamin et al., 2006). From its appearance, we speculate that emu (and potentially cassowary; though note limitations discussed below) patellar tendon morphology might be the result of assimilation of a periarticular fat pad, and if conferring any function at all, might be that of protecting the tendon against bending, similar to an ossified patella (though not conferring its other functions; e.g., mechanical advantage, because it presumably does not increase the moment arm of the muscle force). Through comparative dissections, we have observed that the deep portion of the emu and cassowary patellar tendon corresponds in its location and attachments to the infrapatellar fat pad of other birds, lending some support to our fat pad assimilation hypothesis. If this hypothesis were correct, it may represent a novel solution for dealing with tendon compression. Other animals lacking ossified patellae have patellar tendon modifications (e.g., a fibrocartilaginous patelloid structure in marsupials (Reese et al., 2001)) which are thought to play a similar protective role. However, so far emus (and possibly cassowaries) appear unique in possessing diffuse cartilaginous changes and incorporation of fat within the patellar tendon.

Note that fewer inferences can be made for the cassowary than for emus, as we were limited in this study to soft tissue observations from a single skeletally mature individual. Given that cassowaries and emus are closely related (Fig. 1), it is not surprising that they appear to have similar patellar tendon morphologies: i.e., fatty and collagenous with some cartilage-like tissue, without a patellar ossification, and distally split into deep and superficial portions.

We note that, ideally, samples destined for histological examination would be immediately snap-frozen to facilitate detailed immunohistochemical identification of cell types/tissues. The use of slow-frozen specimens with various and often unknown timings is far from desirable. However, it is somewhat necessitated in studies such as ours because the animals often perish in places without facilities for rapid freezing, are rare and crop up infrequently for scientific use, and are often stored for variable lengths of time until enough are amassed for study.

In attempting to complete our character matrix, which had ambiguous entries due to a lack of published data, we have observed osseous patellae in several tinamou and kiwi specimens. Some previous studies note the patella to be a cartilage (or fibrocartilage) structure in kiwis and tinamous (Owen, 1840; Parker, 1864), which has led to it being considered absent as a skeletal feature by others (e.g., for tinamous in Livezey & Zusi, 2006). In other birds, the patella begins as a cartilaginous nodule and ossifies late relative to the other bones (Lansdown, 1970; Hogg, 1980; Pourlis & Antonopoulos, 2013). Considering our studies of museum specimens it seems likely that this is also the case for (at least some species of) kiwis and tinamous. It is also possible that in these species, the patella does not ossify in every mature individual (e.g., the apparent absence of a patella in Beale’s (1985) and Beale’s (1991) studies of kiwis). Other sesamoid bones can exhibit variable presence between individuals (e.g., dorsal carpal sesamoid (Hogg, 1980)); however this would be unusual for the patella which, as far as research to date has shown, is a relatively constant feature in the species possessing it. Much larger sample sizes of mature individuals would be needed to answer this question.

Unfortunately, the lack of patellae in other palaeognath specimens cannot reliably be used to infer their absence. As small, late-ossifying bones buried in soft tissue, accidental loss would also be a very plausible explanation for the lack of patellae in osteological museum specimens. For the species where patellar presence and form remains unclear, further study of tissues in situ is necessary, or substantial evidence from well-preserved museum or fossil specimens with an unambiguous ossified patella in situ.

The situation with moa specimens poses an excellent example. Although Owen (1883) figured and described what he thought to be a patella in one Dinornis specimen, we consider this identification dubious. The absence of a patella in so many other moa skeletons (>50 specimens of reasonable completeness/association; taxonomy in some disarray but including numerous Dinornis and Pachyornis; also Anomalopteryx, Emeus, Euryapteryx, Genyornis) examined in the NHMUK collection and in the literature, despite the preservation of gut contents, scleral ossicles, tracheal rings and other fine details in the same specimens, renders the argument that the patella simply was not preserved or found to be implausible. Unfortunately, Owen’s single “patella” specimen has gone missing in the NHMUK collections (S Chapman, pers. comm., 2014). It is likely to have been a tarsal sesamoid or other bone rather than a patella, as the tarsal sesamoids are grossly similar in size and shape to a patella and are commonly present in moa skeletons (our observations of numerous NHMUK specimens of Dinornis and Pachyornis), and yet Owen did not describe a tarsal sesamoid. Regardless, future studies of the anatomy of diverse dinornithiform (and other Palaeognathae, including the key stem taxa called lithornithids) species should remain alert for the possibility of the presence of a patella in some species or individuals, to test our assumption and Owen’s identification and to refine understanding of the evolution of the patellar sesamoid in Palaeognathae.

Considering that the patella may have played an important role in the locomotor adaptation of palaeognaths, elucidation and comparison of its morphology may also prove informative in the question of whether ratite flightlessness was inherited from a single flightless ancestor (synapomorphic) or whether flight has been convergently lost on multiple occasions in ratites (e.g., Harshman et al., 2008; Johnston, 2011; Smith, Braun & Kimball, 2013; Baker et al., 2014; Mitchell et al., 2014). Though a conclusion cannot be made on the basis of this one piece of evidence, significant differences in ratite knee anatomy and patellar presence/absence are more plausible in a scenario where there are repeated losses of flight and convergent evolution of cursorial specialisations.

Curiously, the loss(es) of an ossified patella in palaeognaths seems paralleled by other reductions (or plesiomorphic absences?) of accessory ossifications in the limbs: the tarsal sesamoid seems lacking in most taxa except some moa (see above; present in numerous NHMUK specimens of Pachyornis and Dinornis), and ossifications of the lower limb tendons are likewise absent in almost all taxa—although we have noticed them in some tinamou specimens, and Houde & Haubold (1987) described them in the stem ratite Palaeotis. Both tarsal sesamoids and ossified digital flexor tendons were probably ancestrally present in the most recent common ancestor of crown group birds (e.g., Hogg, 1980; Hutchinson, 2002), although even less is known about the tarsal sesamoid than the patella in birds. The coincidence of these reductions of limb ossifications with the evolution of cursorial morphology, flightlessness and large body size in Palaeognathae remains an intriguing mystery.

Our reconstructions of patellar evolution support the inference that the patella has a single origin in the distant ancestor of modern birds (the clade containing Hesperornithiformes + Neornithes), rather than independent evolutions in Palaeognathae and Neognathae. Given our results, the form of the ossified patella in their common ancestor is most likely to have been a small, flattened flake of bone rather than a large, lever-like structure as in later birds. As might be expected, the different tree topologies proposed for Palaeognathae result in different patterns of patellar evolution within this clade. However, we infer that within palaeognaths, the osseous patella has been lost at least once by emus and cassowaries, and likely in moa too, possibly separately (Fig. 9). In contrast, an ossified patella has been retained by some other palaeognaths and even evolved into a larger structure (e.g., ostriches; Chadwick et al., 2014). Data from other species (cassowaries, rheas, and especially fossil taxa such as Lithornis, stem ostriches or hesperornithiforms, or less well preserved taxa such as Ichthyornis) as well as a robust palaeognath phylogeny would be of great value in more clearly tracing the evolution of the patella in this clade.

Interestingly, the hypothesised origin of the patella coincides with the evolution of knee-flexion-based hindlimb kinematics seen in modern birds (as opposed to hip-extension seen ancestrally) in the late Jurassic/early Cretaceous, according to Chiappe (1995), Hutchinson (2002) and others. This could be interpreted as corroboration of the patella’s presumed protective/biomechanical role(s)—perhaps the patella formed as a consequence of new stresses experienced by the hindlimb in its modern configuration (as a type of evolutionary “spandrel” that has become genetically fixed), or perhaps its presumed biomechanical advantages facilitated the acquisition of new locomotory styles? Much further research is needed to begin addressing these possibilities and other questions raised by our study, such as why some birds (e.g., emus) have apparently lost the patella whilst others with broadly similar locomotion (e.g., ostriches) have developed large patellar sesamoids.

We thank Luis Lamas, Kyle Chadwick, Emily Sparkes, Kayleigh Rose and Charlotte Brassey for specimen acquisition, and for access and use of CT images (Luis Lamas for emus, Kyle Chadwick for ostrich, and Charlotte Brassey for cassowary). Thanks to Vivian Allen for help with Fig. 2A and Kyle Chadwick for usage of his 3D model in Fig. 8B. Thanks also to Andrea Pollard, Alexander Stoll, Ken Smith, Medina Shanahan and George Fries for their help with the histological aspects of this study. Finally, many thanks to Trevor Worthy and an anonymous reviewer for helpful comments on a previous version of this manuscript.

Additional Information and Declarations

Competing Interests

Author Contributions

Animal Ethics

John R. Hutchinson is an Academic Editor for PeerJ.

Sophie Regnault performed the experiments, analyzed the data, wrote the paper, prepared figures and/or tables.

Andrew A. Pitsillides analyzed the data, contributed reagents/materials/analysis tools, reviewed drafts of the paper.

John R. Hutchinson conceived and designed the experiments, analyzed the data, reviewed drafts of the paper.

The following information was supplied relating to ethical approvals (i.e., approving body and any reference numbers):

Only cadaver material was used in this study; no ethical approval was required (the animals were not killed for this study).

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
