# Peer review of "Structure, ontogeny and evolution of the patellar tendon in emus (Dromaius novaehollandiae) and other palaeognath birds"

_PeerJ, doi:10.7717/peerj.711_

## Round 0.1 · original submission · Major Revisions

I invite you to resubmit this manuscript after you have addressed all of the issues raised by the two reviewers.

·

Basic reporting

This manuscript was a pleasure to read as it was very well written, to the extent it needs no editing. The arguments are well put and the gathered data pertinent and interpreted justifiably. The information is novel and a useful contribution to knwledge.
I can think of no substantive suggestion re content or anything else that would improve it.

Experimental design

As above the design of this project is excellent and its implementation also excellent

Validity of the findings

I can think of no way that this could have been improved, with one exception. If the development or ossification of the patella is related to age as some data suggested, the authors could have made a statement about the general ontogeny of emus, ie something along the lines that they reach full size within about 9 months and appear osteologically mature by the end of 1 year (this unlike the kiwi) thus a ossified patella should appear by 18 months. But would be nice to have data for say a mature adult at about 3 years plus.

Additional comments

I noted a couple of issues re the references: Hackett et al has 18 authors - about half have been omitted.
Macalister 1864 - surely the JSTOR in this ref is part of the medium by which the article was obtained - it cannot be the reference - so a volume and part is missing.

I suggest that the authors check with Duncan 1937 regarding the correct dates for the Trans Zool Soc of Lond. and the Proceediungs - this is important as the online version gives the incorrect dates IN MANY INSTANCES - ie it gives the year they were for - NOT the year that they were published.
Duncan, F. M. 1937. On the dates of publication of the Society’s ‘Proceedings’, 1859-1926. With an Appendix containing the dates of publication of ‘Proceedings’ 1830-1858, compiled by the late F. H. Waterhouse, and the ‘Transactions’, 1833-1869, by the late Henry Peavot, originally published in P. Z. S. 1893, 1913. Proceedings of the Zoological Society of London 107: 71-84.

For example. 'Owen P. 1839. On the anatomy of the southern Apteryx (Apteryx australis, Shaw). The Transactions of the Zoological Society of London 2:257-301.'
was published on 6 April 1840 so is incorrectly listed as 1839.
However Owen 1883 and Parker 1864 appear correct.

Pearson and Davies 1921a - delete the a.

Trevor Worthy

Reviewer 2 ·

Basic reporting

Introduction:
- Requires additional references (some statements are left uncited, e.g., lines 20-24), organization, and clarification (see comments below).
- Lines 20-24: Why is the basal position of palaeognathous birds (flightless) important to the evolution of the patella in all other crown clade birds? Does this tell us something about the evolution of flight? As written, it leaves too much to be inferred by the reader.
- A basic, agreed upon phylogenetic tree right at the start would provide graphic information for clarity, as well as to put the clade into a frame of reference for the phylogenies presented later.
- Lines 28 to 31, it is stated: "The patella’s potential to alter limb mechanics means that elucidation of this tendon’s configuration could provide insight into the evolution of cursoriality and large body size in ratites, and whether flight has been lost multiple times in palaeognaths". The locomotion of birds, ratites in particular, and biomechanics of the patella needs to be developed further. Thus, the introduction fails to describe the historical discussion of the biological functions of the patella across animals in order to set up why it is important to understand the evolution of the patella in birds, specifically emus (see comments below). A discussion of the various functional hypotheses of the patella, as well as the similarities and differences of the patella in other tetrapods is required for basic understanding of the topic by the reader.

Methods:
- What is the skeletal maturity of an emu? What does 18 months (and the other time points) equate to in skeletal age of an emu? Is this comparable to other ratites and birds in general?
- In the absence of snap-freezing, a slow rate of freezing to -20 is very damaging to cells and cellular structures for investigation of histological samples. Furthermore, what was the time frame between euthanization and sample freezing? How long were samples frozen before being thawed for collection? Without justification, these methods bring doubt to the accuracy of the results.
- It is unclear why only one adult ostrich and one adult guinea fowl are used for comparative purposes. Requires clarification.
- Line 119-121: Where were these museum specimens obtained from and what specifically about the tendon was being investigated regarding their gross morphology? Requires better organization of the information presented.
-Only staining the ostrich sections and no other samples with Masson’s trichrome needs to be justified.
- Additional information of the museum samples is required: AGE, sample size, sex, collection numbers, etc. for reproducibility. See comments below in experimental design.

Results:
- Figure 1: Parts A through C: The orientation of the specimens should be consistent, they are confusing as shown. Part B and C: Lines 169-176 references to B and C need to be switched, wrong figures indicated. Part G: Why reindeer? This should be ostrich or tinamou to show that they do have a discernible patella for comparison.
- Line 180: von Kossa data should be shown, absence/presence is the basis of the study.
- Tissues described require further staining for definitive identification. See comments in experimental design.
- Table lacks references.
- Figure 6 is not informative of an osseous patella as shown. Requires radiographic imaging to be sufficient.

Discussion:
- Lines 279-281: "Unlike the typical tendon of most tetrapod vertebrates, in which the microanatomy consists predominantly of dense, parallel collagenous bundles" needs to be introduced in the introduction to contrast to emu findings in discussion.
- Lines 333-340: "A commonly presumed function of sesamoids is protective": All functions of sesamoids needs to be described in intro. Why "cushioning tendon and stress dissipation" is concluded from the other tenable hypotheses needs to be justified after backing results up with the appropriate stains.
- Lines 383-384: Confusing in present location. Should be presented above, as previously stated, and conclusions for the hypothesized biological role of the emu patella clearly stated.

Experimental design

- The study lacks critical histological staining for the correct identification of the tissues described. Definitive identification of collagen requires staining by toluidine blue and/or picrosirius red and their birefringence investigated. Toluidine blue will also assist with the identification of chondrocytes and metachromasia. H&E is a very general stain and lacks the specificity the authors attempt to describe. As Thus, the tissues described require further staining to definitively identify collagen, chondrocytes, and to rule out fibrocartilage.
- von Kossa staining was conducted, but not shown. At least one von Kossa image is required to show the findings of this paper revolving around the presence/absence of a patella in emus.
-Alizarin red can also be used in conjunction with von Kossa to clearly demonstrate the absence of a calcified/mineralized patella in emus to support these findings.
- Lacks consistent procedures between emus, ostrich, and guinea fowl
- Is 18 months considered skeletal maturity for an emu? This needs to be justified, otherwise the reader is left wondering if 18 months is old enough.
- How were the museum specimens aged? Actual age of museum specimens is rarely documented. A robust explanation of how these specimens were accurately aged is required.

Validity of the findings

Conclusions are not robustly supported by the evidence presented. Methods of tissue collection for histological sectioning need to be justified. Although it is difficult to conduct statistics with histological studies, sample collection and sample sizes need to be controlled and consistent between age groups and between species. All museum samples need to be thoroughly described and documented. von Kossa results (even though they are negative--still data!!) are not shown, and thus a major part of the study is missing.

---

## Round 0.2 · accepted · Accept

I believe that you have worked very hard to address the concerns of the reviewers and the manuscript is much improved over the original submission.